# Infections in a Tertiary Care Pediatric Center in Southern Poland during the SARS-CoV-2 Pandemic

**Joanna Klepacka [1], Zuzanna Zakrzewska [2], Małgorzata Czogała [2,3]** **, Adriana Chromy-Czoniszwili [4], Szymon Skoczeń [2,3,*]** **and Krzysztof Fyderek [5,6]**

1  Department of Microbiology, University Children's Hospital, 30-663 Krakow, Poland; jklepacka@usdk.pl
2  Department of Oncology and Hematology, University Children's Hospital, 30-663 Krakow, Poland; zuzanna.zakrzewska@doctoral.uj.edu.pl (Z.Z.); malgorzata.czogala@uj.edu.pl (M.C.)
3  Department of Pediatric Oncology and Hematology, Institute of Pediatrics, Jagiellonian University Medical College, 31-008 Krakow, Poland
4  Hospital Pharmacy, University Children's Hospital, 30-663 Krakow, Poland; aczoniszwili@usdk.pl
5  Department of Pediatrics, Gastroenterology and Nutrition, University Children's Hospital, 30-663 Krakow, Poland; krzysztof.fyderek@uj.edu.pl
6  Department of Pediatrics, Gastroenterology and Nutrition, Jagiellonian University Medical College, 31-008 Krakow, Poland
*  Correspondence: szymon.skoczen@uj.edu.pl; Tel.: +48-126-580-261

**Abstract:** The worldwide surge of severe acute respiratory syndrome coronavirus 2 (SARS-CoV-2) has caused a global pandemic and led governments to control spread of the virus and provide care for the population affected by the infection. Although, in children, COVID-19 is usually asymptomatic or mild (except PIMS), the pandemic affected the whole socioeconomic system and led to the overwhelming of healthcare facilities. We report retrospective observations of the prevalence of various infectious diseases during the SARS-CoV-2 pandemic in a tertiary multidisciplinary pediatric center in Southern Poland. We retrospectively evaluated the impact of the SARS-CoV-2 pandemic on the number of other infections diagnosed in a pediatric tertiary care referral center. Our analysis included the period from the beginning of February to the end of April 2020 (spring pandemic wave), and from the beginning of September to the end of November 2020 (autumn pandemic wave). We compared them to the appropriate periods of 2019. The evaluation included blood, urine, stool and lover respiratory tract cultures as well as virological investigations. Additionally, the costs of antibiotics and antifungal drugs in selected departments were assessed. Our analysis showed considerable reduction in the majority of common infections except for influenza A and B. The microbiological data correspond with economical summary of antibiotic costs, which were significantly lower during the pandemic. One exception was the number of positive blood cultures, which increased even though the overall number of tests was lower. A general reduction of the number of infections diagnosed in children could result from the implemented preventative measures associated with the pandemic and the generally increased awareness of the risk of infection among parents and guardians. The treatment of the most serious diseases continued as it did before the pandemic. To our knowledge, this study is the first attempt to assess the impact of the COVID-19 pandemic on the prevalence of infections in a large pediatric center. Further research on the impact of the COVID-19 pandemic on the healthcare systems is necessary.

**Keywords:** common infections; SARS-CoV-2 pandemic

## 1. Introduction

The University Children's Hospital (UCH) is a tertiary care referral center admitting the most severely ill patients, mainly from the southern regions, but also from other parts of Poland. UCH has 469 beds and offers services in all pediatric and surgical specialties (including modern operation rooms and an intensive care department). In 24 departments

of UCH, children from neonates to 18 years old are treated and managed, including those with advanced stages of various neoplastic diseases, burns, congenital malformations, or neonates with extremely low birth weight. Patients with oncological and hematological diseases receive comprehensive treatment in the departments of oncology and hematology, the transplantation center and the department of radiotherapy. The hospital provides full diagnostic and physiotherapy services. The outpatient center has 35 outpatient clinics. The hospital has 33,000 admissions per year and performs approximately 7000 surgical procedures including 450 cardiac operations; more than 170,000 consultations are provided by the outpatient clinic and approximately 34,000 by the emergency department. UCH is also a teaching base for the Institute of Pediatrics and is involved in research and the education of students of medicine, pharmacy, medical analytics, public health, nursing and other medical specialties.

The first information about SARS-CoV-2 in China was acknowledged in the hospital at the beginning of January 2020. At the end of January first testing guidelines were dispatched by Polish National Sanitary Inspection to municipal hospitals, and meetings with municipal authorities were organized. As the signals coming from other countries were rather calming, the Polish Ministry of Health did not push to introduce special precautions at that time. Emerging data from Italy and France urged the Polish authorities to introduce dedicated procedures. Admissions were coordinated and dedicated ICU and COVID departments were organized for patients who could not be hospitalized in the Pediatric Infectious Diseases Department due to clinical conditions. Fortunately, at the beginning of the pandemic, the prevalence of SARS-CoV-2 positivity in Polish population increased relatively slowly, giving time for Ministry of Health and hospitals to improve equipment stocks and establish necessary procedures. On 11 March 2020, the WHO announced the COVID-19 pandemic. Since schools all over Poland were closed on 12 March 2020, the hospital experienced staff shortages because many employees are been mothers of preschool and school children who had to stay home to take care of their children, as provided for by national regulations. Nevertheless, since the beginning of April 2020, the rotation of medical staff has been implemented. In April 2020, regular testing of children before admission to hospital, as well as of medical staff, were introduced. We experienced similar staff shortages in fall 2020, during the second wave of the pandemic in Poland.

In the studied periods, 400 patients with SARS-CoV-2 were hospitalized in the two Pediatric Infectious Diseases Departments in Krakow.

To date (1 April 2021–end-of-day report) 2.36 million SARS-CoV-2 infections were recorded in Poland, including 170,000 in our region. There were 53,665 deaths in Poland.

According to the literature, the morbidity and mortality of SARS-CoV-2 infection was lower in children and the course of the disease was more benign than in adults [1–6]. The burden on pediatric centers less severe than that on adult centers, although post COVID PIMS (pediatric inflammatory multisystem syndrome) started to be an increasing problem. Simultaneously we observed that fewer children sought medical help for other health problems, especially for infectious diseases. The goal of the study was a retrospective observation of confirmed infections in the time of the pandemic.

## 2. Materials and Methods

The impact of changes to the organization of the hospital during the pandemic on the prevalence of infections was analyzed. We retrospectively evaluated the impact of the SARS-CoV-2 pandemic on the numbers of other infections diagnosed in UCH. In the analysis we included the period from the beginning of February 2020 to the end of April 2020, referred to as the spring pandemic period, and compared it to the analogous period of 2019 (referred to as the spring prepandemic period). Moreover, we collected data from the second wave of the pandemic in Poland—from the 1 September to 30 November 2020 (referred to as the autumn pandemic period) and compared it to the analogous period of 2019 (referred to as the autumn prepandemic period). We also compared orders of class

3 antibiotics and antifungal drugs (indicated in the most severe infections) in our hospital in the same periods 2019 and 2020. The precise definition of class 3 antibiotics was as follows: antibacterial and antifungal drugs prepared by the hospital pharmacy in individual doses (like linezolid, teikoplanin, levofloxacin, ertapenem, micafungin, caspofungin, amfotericin B, voriconazole). Due to high cost of those drugs, it was optimal from the economic point of view to introduce a unit–dose system.

The data were collected directly from the Department of Microbiology of UCH and the hospital pharmacy. Basic statistical analyses including Pearson's chi-squared test were performed using STATISTICA 13 software (StatSoft, Tulsa, OK, USA).

## 3. Results

From 1 February to 30 April 2019, the microbiology laboratory performed 2725 blood cultures, of which 181 (6.6%) were positive. In the analogous period of 2020, the number of blood cultures performed was 2115, of which 211 (10%) were positive. From the beginning of September to the end of November 2019, the number of blood cultures performed was 2701, of which 267 (9.89%) were positive. In fall 2020, the number of blood cultures performed was 2409, of which 178 (7.39%) were positive. (Figure 1, Table 1).

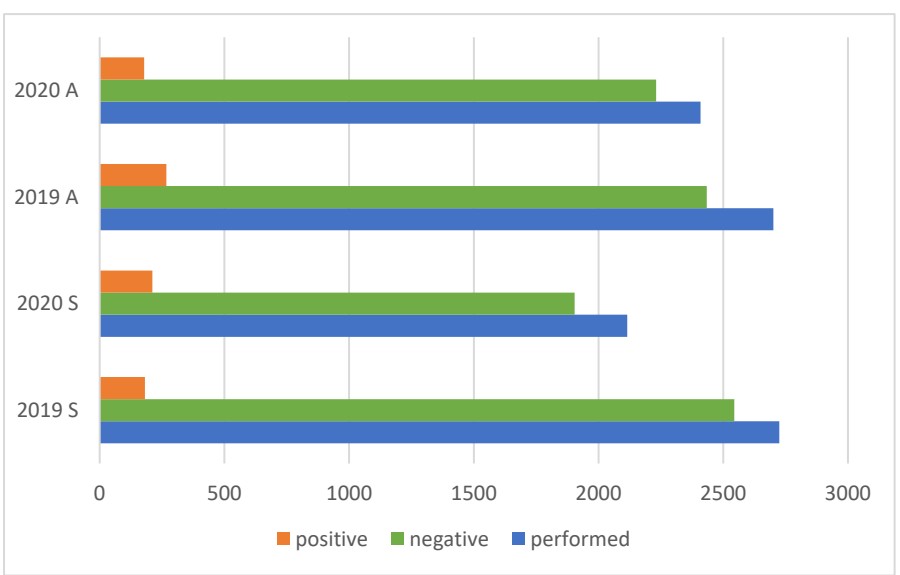

**Figure 1.** Blood cultures.

**Table 1.** Blood cultures.

| Period | Performed | Negative | Positive | |
|--------|-----------|----------|----------|--------|
| 2019 S | 2725 | 2544 | 181 | *p* < 0.0001 |
| 2020 S | 2115 | 1904 | 211 | |
| 2019 A | 2701 | 2434 | 267 | *p* = 0.0016 |
| 2020 A | 2409 | 2231 | 178 | |

The overall difference between the numbers of blood cultures performed in 2019 and 2020 was 902 samples (17%).

The percentage of positive blood cultures in the pandemic periods increased significantly: by 14% (181 vs. 211 samples) (*p* < 0.0001) in the spring periods, and by 33% (178 vs. 267 samples) (*p* = 0.0016) in the fall periods.

The pathogen most frequently isolated from blood samples was *Staphylococcus epidermidis*. It was found in 73% of positive samples in 2019 and 75% in 2020 (Figure 2, Table 2). We found a decreasing number of coagulase-negative staphylococci and *Pseudomonas aeruginosa* in the pandemic periods, despite the higher overall numbers of positive blood cultures.

Differences were not statistically significant (Table 2). The number of positive cultures in Figures 1 and 2 were different because some studies were repeated in the same patients.

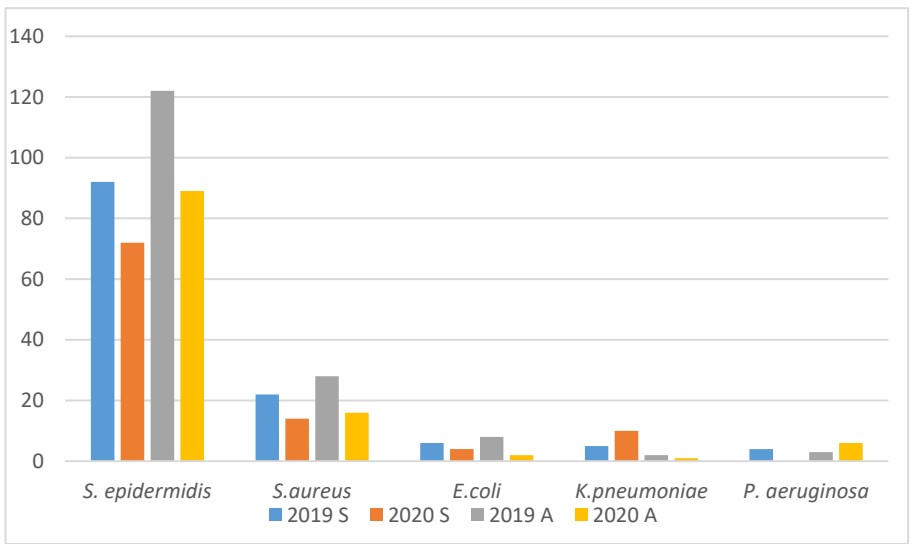

**Figure 2.** Pathogens found in blood cultures.

**Table 2.** Pathogens found in blood cultures.

| Pathogen | 2019 S | 2020 S | 2019 A | 2020 A |
|---|---|---|---|---|
| *S. epidermidis* | 92 | 72 | 122 | 89 |
| *S. aureus* | 22 | 14 | 28 | 16 |
| *E. coli* | 6 | 4 | 8 | 2 |
| *K. pneumoniae* | 5 | 10 | 2 | 1 |
| *P. aeruginosa* | 4 | 0 | 3 | 6 |
| | *p* = 0.1515 | | *p* = 0.3029 | |

The overall number of urine cultures decreased (Figure 3, Table 3). No considerable differences were found in the profiles of microorganisms isolated from urine samples in the studied periods. The predominant pathogen was still *Escherichia coli*. The distribution of the detected pathogens was similar. The prevalence of *Enterococcus* spp. (*Enterococcus faecium* and *Enterococcus faecalis*) was lower (Figure 4, Table 4).

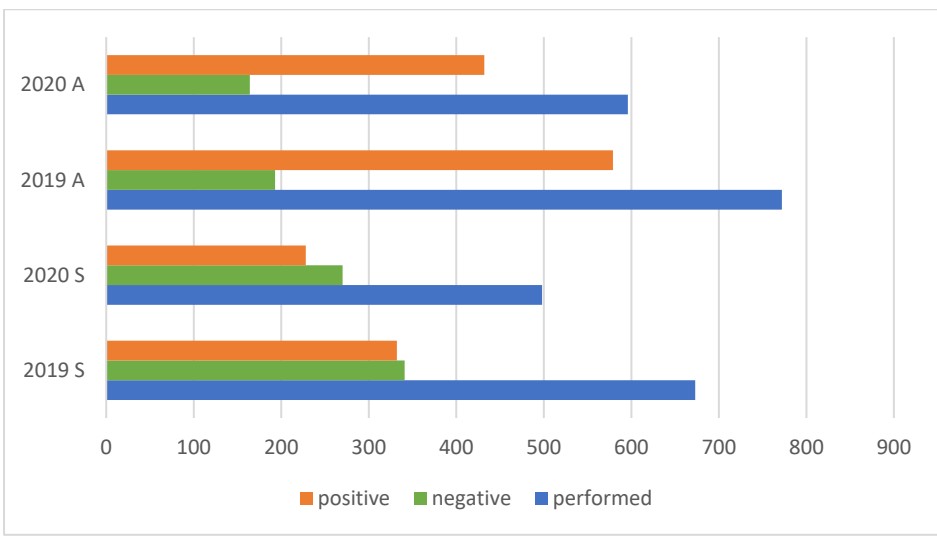

**Figure 3.** Urine cultures.

**Table 3.** Urine cultures.

| Period | Performed | Negative | Positive | |
|--------|-----------|----------|----------|---|
| 2019 S | 673 | 341 | 332 | *p* = 0.2295 |
| 2020 S | 498 | 270 | 228 | |
| 2019 A | 772 | 193 | 579 | *p* = 0.2933 |
| 2020 A | 596 | 164 | 432 | |

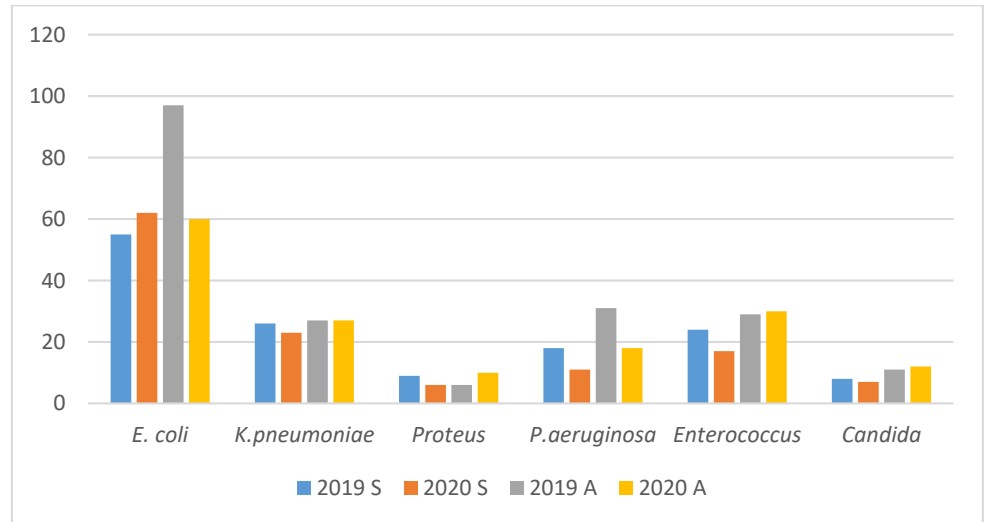

**Figure 4.** Pathogens found in urine cultures.

**Table 4.** Pathogens found in urine cultures.

| Pathogen | 2019 S | 2020 S | 2019 A | 2020 A |
|----------|--------|--------|--------|--------|
| *E. coli* | 55 | 62 | 97 | 60 |
| *K. pneumoniae* | 26 | 23 | 27 | 27 |
| *Proteus* | 9 | 6 | 6 | 10 |
| *P. aeruginosa* | 18 | 11 | 31 | 18 |
| *Enterococcus* | 24 | 17 | 29 | 30 |
| *Candida* | 8 | 7 | 11 | 12 |
| | *p* = 0.6345 | | *p* = 0.1595 | |

Additionally, we compared profiles of microorganisms detected in samples from lower respiratory tract (Figure 5, Table 5). In 2019, 356 samples were tested, compared to 268 in 2020. We found significant reductions in isolated *Candida* species, from 19 in 2019 to only 2 in 2020 (*p* = 0.0031). No significant differences with respect to *E. coli* and *Klebsiella pneumoniae* were found. A decline of *Staphylococcus aureus* and reduced numbers of *P. aeruginosa* and *Haemophilus influenzae* isolates from the lower respiratory tract was observed.

**Table 5.** Bronchoalveolar lavage fluid cultures.

| Pathogen | 2019 S (356) | 2020 S (268) |
|----------|--------------|--------------|
| *Candida* | 19 | 2 |
| *E. coli* | 17 | 16 |
| *K. pneumoniae* | 25 | 22 |
| *P. aeruginosa* | 27 | 5 |
| *S. aureus* | 38 | 21 |
| *H. influenzae* | 15 | 5 |
| | *p* = 0.0031 | |

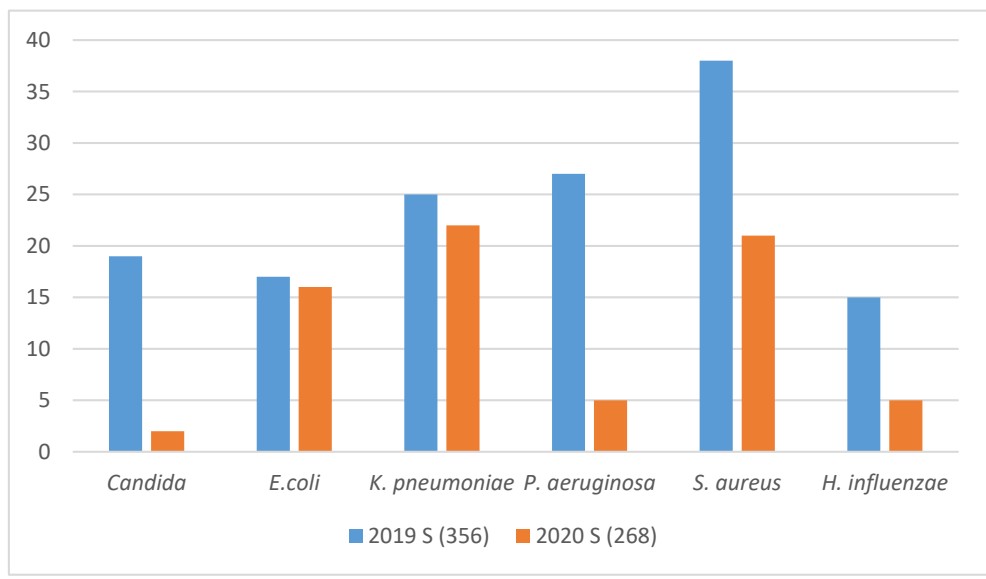

**Figure 5.** Bronchoalveolar lavage fluid cultures.

The overall number of ordered bacterial and fungal stool cultures decreased by 389 (14.3%) compared with both periods: in 2019 the number of stool cultures performed was 2726, and in 2020 it was only 2337 (Figure 6). The number of fungal cultures performed in 2020 was 77 (24.8%) lower than in 2019. Moreover, there were 52 fewer positive cultures than in 2019; the difference was not statistically significant (Table 6, Figure 7). There were no significant differences in the profiles of *Candida* spp. isolates (Figure 8).

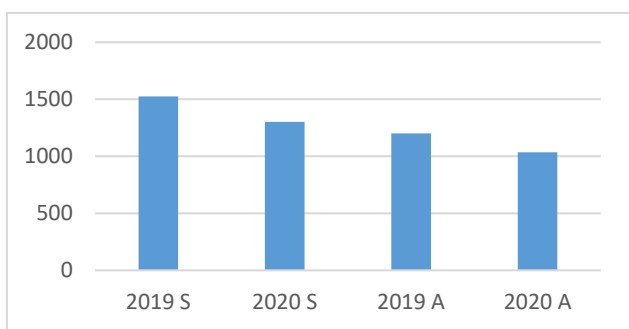

**Figure 6.** Stool cultures.

**Table 6.** Mycological stool cultures.

|        | Performed | Positive | Negative |              |
|--------|-----------|----------|----------|--------------|
| 2019 S | 311       | 171      | 140      | *p* = 0.2104 |
| 2020 S | 234       | 116      | 118      |              |

We also assessed the differences in pneumotropic viruses detected in throat swabs (Figures 9 and 10). Reduced numbers of performed RSV tests as well as of positive results were seen. In 2019, the results of 27% of the tests were positive, compared with 32% in 2020. We found 42 (43%) fewer RSV-positive infections in 2020 than in the analogous period of 2019, but the difference was not statistically significant (Table 7). We confirmed 82 cases of influenza A and B in 2019, which was of 18% the tests performed. In comparison, there were only 52 cases in 2020 (27% of performed tests were positive), the difference was not statistically significant (Table 8).

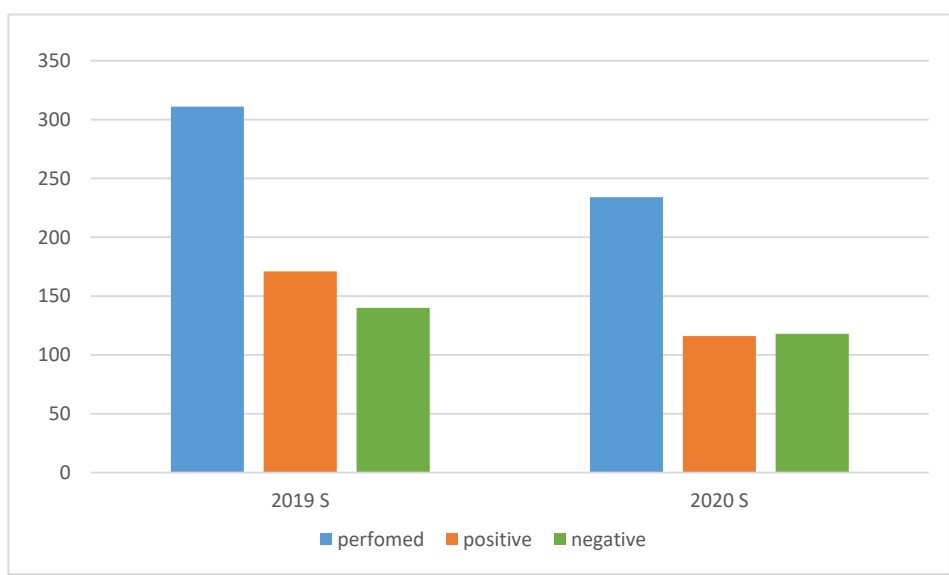

**Figure 7.** Mycological stool cultures.

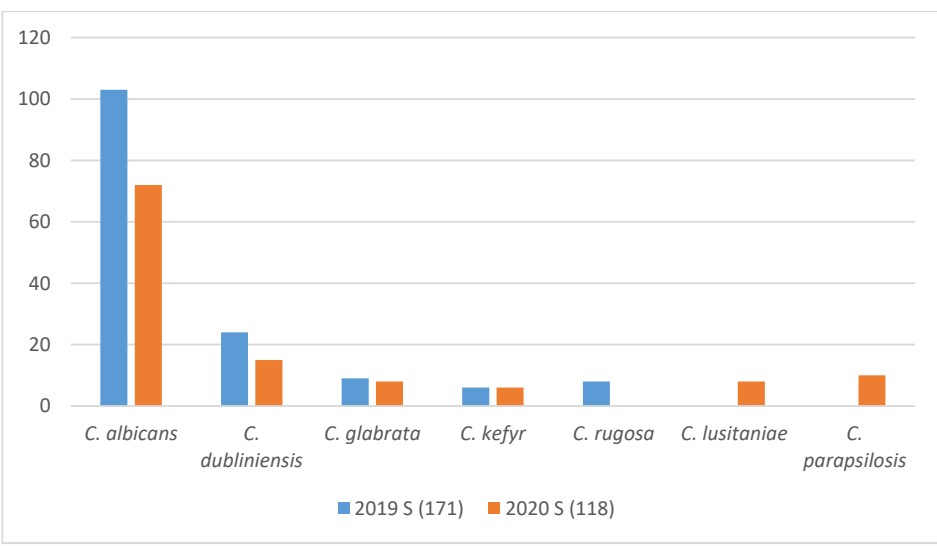

**Figure 8.** *Candida* species isolated from stool samples.

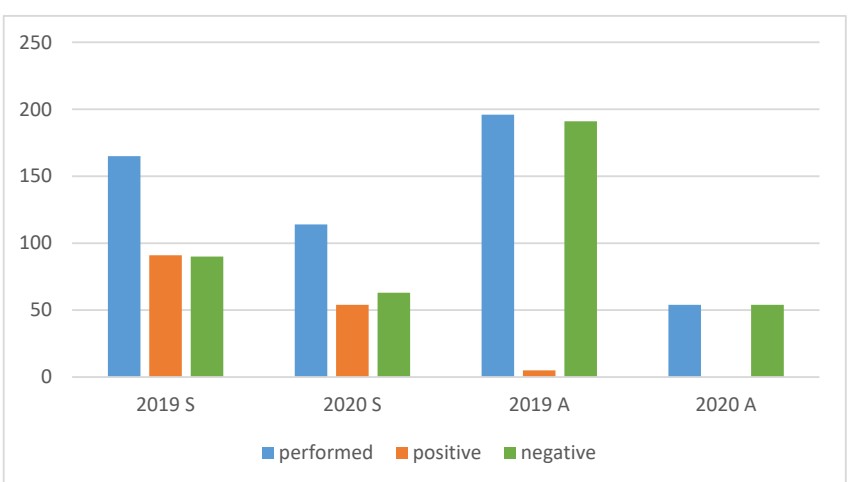

**Figure 9.** Comparison of RSV tests.

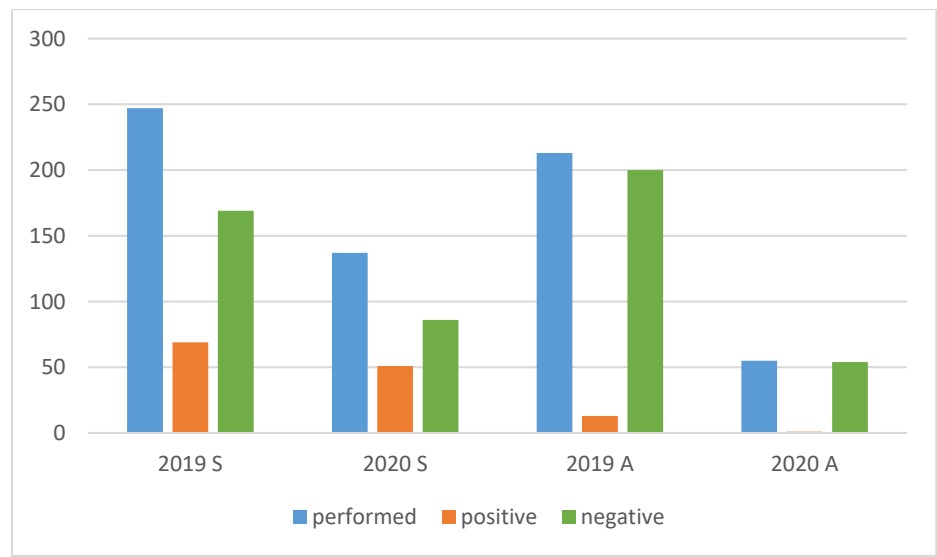

**Figure 10.** Comparison of Influenza A and B tests.

**Table 7.** Comparison of RSV tests.

| Period | Performed | Positive | Negative | |
|--------|-----------|----------|----------|--|
| 2019 S | 165 | 91 | 90 | *p* = 0.4869 |
| 2020 S | 114 | 54 | 63 | |
| 2019 A | 196 | 5 | 191 | *p* = 0.5243 |
| 2020 A | 54 | 0 | 54 | |

**Table 8.** Comparison of Influenza A and B tests.

| Period | Performed | Positive | Negative | |
|--------|-----------|----------|----------|--|
| 2019 S | 247 | 69 | 169 | *p* = 0.0998 |
| 2020 S | 137 | 51 | 86 | |
| 2019 A | 213 | 13 | 200 | *p* = 0.3506 |
| 2020 A | 55 | 1 | 54 | |

We found a radical decline of rotavirus infections in the hospital (Table 9, Figure 11). In the spring of 2020, we performed 110 fewer rotavirus tests and received 51 fewer positive results (*p* = 0.0172). In fall 2020, we conducted 158 fewer rotavirus tests and received 12 fewer positive results (*p* = 0.6705).

**Table 9.** Comparison of rotavirus tests.

| Period | Performed | Positive | Negative | |
|--------|-----------|----------|----------|--|
| 2019 S | 763 | 177 | 584 | *p* = 0.0172 |
| 2020 S | 488 | 86 | 402 | |
| 2019 A | 585 | 31 | 554 | *p* = 0.6705 |
| 2020 A | 343 | 16 | 327 | |

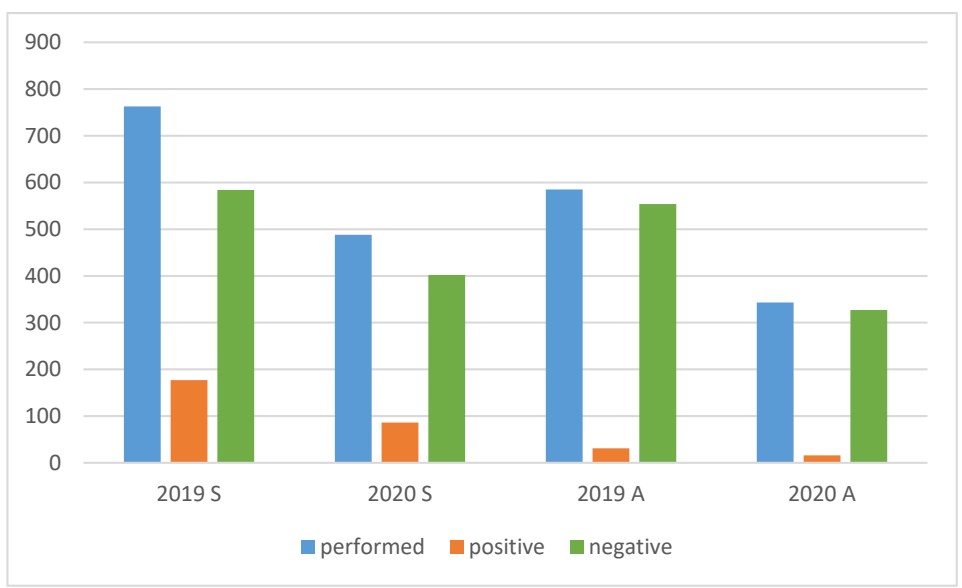

**Figure 11.** Comparison of rotavirus tests.

The costs (based on current exchange rates) of all groups of antibiotics utilized in UCH showed a 29% decrease (February/March/April 2019 vs. 2020). However, when it came to the same periods in autumn, no differences were seen (Table 10). We also put the const of antifungal drugs in the analysis. The orders for class 3 antibiotics were closely analyzed as they were prescribed in the most severe infections. The class 3 antibiotics costs assigned to respective departments of the hospital are presented in Table 11.

**Table 10.** Costs of antibiotics by category (PLN) (average exchange rate in 2020: 1 USD = 3.8972 PLN).

| Drug Class | Spring 2019 | Spring 2020 | Autumn 2019 | Autumn 2020 |
|---|---|---|---|---|
| Antibiotics class I | 96,508 | 78,655 | 100,560 | 90,539 |
| Antibiotics class II | 148,955 | 92,232 | 123,768 | 137,964 |
| Antibiotics class III | 26,474 | 21,155 | 28,078 | 23,629 |
| Summary | 271,937 | 192,043 | 252,406 | 252,132 |
| Antifungal | 337,410 | 119,558 | 102,184 | 397,207 |

**Table 11.** Costs of class 3 antibiotics by hospital department (PLN) (average exchange rate in 2020: 1 USD = 3.8972 PLN).

| Department | 2019 A | 2020 A | 2019 S | 2020 S |
|---|---|---|---|---|
| intensive care unit | 11,797.06 | 12,640.5 | 10,020.29 | 7020.57 |
| surgery | 5857.83 | 5500.91 | 4033.37 | 1882.58 |
| cardiology | 264.38 | 0 | 283.56 | 0 |
| oncology/hematology | 3903.12 | 3253.82 | 4423.68 | 3483.28 |
| STEM cell transplantation center | 2020.9 | 0 | 4651.07 | 3678.63 |
| neonatal intensive care unit | 450.15 | 414.72 | 281.34 | 480.49 |
| pulmonology | 362.88 | 1378.94 | 1873.84 | 1390.4 |
| emergency department | 311.04 | 0 | 16 | 34 |
| gastrology | 0 | 0 | 406 | 0 |
| nephrology | 0 | 429.84 | 1244.16 | 345.6 |
| orthopedics | 3110.4 | 0 | 124 | 1996.2 |
| total | 28,077.76 | 23,618.73 | 2735.31 | 20,311.75 |

## 4. Discussion

The current pandemic of the coronavirus disease (COVID-19) dramatically impaired the functioning of the hospital, even though we did not experience any severe outbreak of SARS-CoV-2 infection. For several months, admissions were restricted and staff was reduced. We implemented guidelines and recommendations regarding the management of COVID-19 [5,7,8]. Patients admitted to our center were screened for SARS-CoV-2. We applied social distancing and enhanced hygiene precautions. This situation was reflected by reduced overall numbers of infections detected and also by a decrease in the use of antimicrobial drugs [9]. It was commonly known that some children that should seek medical attention did not appear in hospitals due to the parents' concerns about COVID-19 or being in quarantine. On the other hand, the enhanced hygiene procedures and isolation at home might have reduced the spread of infectious diseases [10,11]. Cardiac surgery, orthopedics, oncology, oncology/hematology and emergency departments ensured the continuum of care for children despite shortages in staff and medical supplies. They did not cancel appointments nor delay admissions for patients treated for life-threatening conditions. Recent investigations have determined that SARS-CoV-2 infection was generally more prevalent in adults than in children, although the youngest children and those with comorbidities are vulnerable to severe COVID-19 [12,13]. Furthermore, the signs and symptoms of COVID-19 in children are less well defined, thus creating particular challenges for medical professionals [14].

The presented data reveal considerable reductions in the most common bacterial infections in the pediatric population treated in one large center at the time of SARS-CoV-2 pandemic. The microbiological data are consistent with the summary of the costs of antimicrobial drugs. One exception was the number of positive blood cultures, which increased even though the overall number of the tests was lower. These results were surprising because the number of performed surgical interventions and therefore the number of potential surgical site infections decreased. We also observed significant reductions in *P. aeruginosa* and coagulase-negative Staphylococci as the result of isolation, more frequent and meticulous disinfection, strict aseptic conditions and precautions during blood sample collection.

The reduction in the number of urine cultures was a result of a decrease in orders from outpatient clinics as well as of routine screenings on admission. The decrease in the numbers of cultures of lower respiratory tracts in 2020 could also result from a temporary closure of pulmonology and intensive care departments which caused difficulties in obtaining appropriate samples. As pulmonology and gastroenterology departments were closed, we observed changes in microbial profiles—a decline of the numbers of *S. aureus*, *H. influenzae*, and *P. aeruginosa* isolates. However, the changes might also be an effect of intensified hygiene procedures or the isolation of the youngest children at home.

The number of stool cultures decreased due to the admission restrictions (routine stool screening). The reduction in positive stool cultures was seen as a result of reduced number of patients hospitalized in transplantation and oncology departments, limited number of overall hospitalized patients and particularly due to a decrease in the numbers of patients treated with broad-spectrum antibiotics.

The prevalence of RSV infections was lower due to the isolation of the youngest children at home. The same trend was observed for rotavirus gastroenteritis, with a significant reduction in infections during the pandemic as a result of strict hygiene conditions and the isolation of infants at home. Interestingly, despite of all precautions, the number of confirmed influenza A and B infections in 2019 and 2020 was similar.

The decreased costs of each category of antibiotics was parallel to the overall numbers of bacterial infections. We closely assessed the orders for class 3 antibiotics as they were prescribed in the most severe, life-threatening bacterial infections. In all departments, reductions of class 3 antibiotic costs were seen, with the exceptions of cardiac surgery, orthopedics, oncology/hematology and emergency departments. These departments were constantly open during the pandemic and admitted patients with the most severe bacterial

infections. Interestingly, we did not observe any differences between the same periods in the fall.

Our results provide evidence that the global spread of SARS-CoV-2 and the implemented preventive measures may have caused reductions in common childhood infections or increases of self-medication. Simultaneously, the numbers of the most severe infections treated in our hospital remained on the same level.

The presented results correspond with data from large pediatric centers in Italy and the US in terms of challenges posed by COVID-19 pandemic [15–17]. These reports concentrated on the identification of patients with SARS-CoV-2 infection and the prevention of disease spread. Unfortunately, there is a knowledge gap regarding the prevalence of common infections in children during the pandemic. There is no adequate data on the patients who might not have received medical help on time due to the pandemic.

To our knowledge, this study is the first to examine an impact of the COVID-19 pandemic on the prevalence of bacterial and fungal infections in children. Our study focused on infection epidemiology mainly from a microbiological point of view. This could be regarded a limitation as we cannot be certain whether in certain patients, we have observed an actual infection or just colonization. Moreover, we did not include information about SARS-CoV-2 test result in our patients. The data come from a single center, although this was a large, tertiary care pediatric referral hospital. We assessed the prevalence of positive microbiological tests and the costs of therapies, but not precise prevalence of infections in the hospital. Despite the limitations, we consider the presented data useful and, in our opinion, this is a good introduction to further studies, as it is one of the first studies presenting data on common infection rates in children during COVID-19 pandemic.

We recommend further research considering the impact of COVID-19 on the entire healthcare system.

**Author Contributions:** Data curation, J.K.; Methodology, M.C.; resources, J.K. and A.C.-C.; wupervision, K.F.; writing—original draft, Z.Z.; writing—review and editing, S.S. All authors approved the final manuscript as submitted and agree to be accountable for all aspects of the work. All authors have read and agreed to the published version of the manuscript.

**Funding:** No funding was secured for this study.

**Institutional Review Board Statement:** Not Applicable.

**Informed Consent Statement:** Not Applicable.

**Data Availability Statement:** Not Applicable.

**Acknowledgments:** The research was conducted on the initiative of the Hospital Committee on Antibiotic Therapy.

**Conflicts of Interest:** Authors declare no conflict of interests.

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
