# Peer review of "Infections in a Tertiary Care Pediatric Center in Southern Poland during the SARS-CoV-2 Pandemic"

_2036-7481, doi:10.3390/microbiolres12040069_

Round 1

Reviewer 1 Report

Klepacka et al. offer an intriguing perspective on the patterns of infections recorded in a reference Pediatric Center in Southern Poland before and during the COVID-19 pandemic. The paper's main strength is its collection of a large dataset of reported infections and subsequent comparison of these infections before and during the COVID-19 pandemic period. This gives the results and interpretations greater credibility. However, the paper is limited by its poor usage of statistical methods to show the significant differences among the values.

Major comments:

-To make any conclusion, the authors should, however, employ proper statistical methods to prove that the differences noticed by the authors are significant differences. Therefore, any significant difference should be followed by p-values throughout the manuscript.

-I do not understand the lines 113-114, “Number of positive blood cultures in pandemic periods increased significantly: 14% (p<0.0001) in the spring periods and by 33% (p=0,0016) in the autumn periods.”  How the authors calculated 14% and 33%?

-The authors should show Influenza A and B infections separately in Figure 10, and then highlight their significance in relation to RSV, and SARS-CoV-2.

Minor comments:

-many grammatical mistakes do exist frequently; consequently, I recommend that authors proofread their papers thoroughly.

-use only standard abbreviations in the paper. For example, SARS-CoV-2 (not SARS-Cov-2), years old (not y.o.)

-describe any abbreviation on its first appearance (such as PIMS)

Author Response

Thank you for your suggestions regarding our manuscript. 

  1. To make any conclusion, the authors should, however, employ proper statistical methods to prove that the differences noticed by the authors are significant differences. Therefore, any significant difference should be followed by p-values throughout the manuscript.   

Answer: The raw data (number of probes) with p-values are shown in tables 1-11. Regarding type of data (epidemiological) we employed chi-squared test and chi-squared test with Yates modification to show changes between pandemic and prepandemic periods.

  1. I do not understand the lines 113-114, “Number of positive blood cultures in pandemic periods increased significantly: 14% (p<0.0001) in the spring periods and by 33% (p=0,0016) in the autumn periods.”  How the authors calculated 14% and 33%?

Answer: We changed the text and added raw data (number of probes): „Percentage of positive blood cultures in pandemic periods increased significantly: by 14%- (181 vs 211 probes) (p<0.0001) in the spring periods and by 33% (178 vs 267 probes) (p=0,0016) in the autumn periods.”

  1. The authors should show Influenza A and B infections separately in Figure 10, and then highlight their significance in relation to RSV, and SARS-CoV-2

Answer: Unfortunately, we don't have exact influenza tests results- we don't distinguish between type A and B. We didn't analyse SARS-CoV-2 positivity in our patients. The study shows changes caused by epidemiological situation.

  1. Minor comments. Answer: we changed the text accordingly.

Reviewer 2 Report

  1. Regarding ethical point of view, did this study submit ethical committee and get certificate although it was the secondary data analysis, the authors should get agreement from Medical superintendent and ethical committee.
  2. We noted the authors need to correct the scientific names of bacteria (E coli, Psudomonas, etc) as Italic form at many palces.
  3. This study highlighted that the number of requested blood culture tests were reduced but the positivity rates were increased. Is there any association with COVID positive patients and blood culture positive patients? Please add the possible reasons for increasing the culture tests positive at the revised manuscript.
  4. Did the authors notice co-infection cases with COVID-19 and other bacterial or viral infection cases? It will be more interesting if the authors can show co-infection cases. Please review and add at the revised one.

5.The authors also described that the COVID-19 infection make great impact on adult patients than paediatrc cases. Why did the authors choose paediatric center for conducting this study?  The study from adult center will be more interesting and add the justification for choosing paediatric center at the revised manuscript.

Author Response

Thank you for comments regarding our manuscript.

  1. Regarding ethical point of view, did this study submit ethical committee and get certificate although it was the secondary data analysis, the authors should get agreement from Medical superintendent and ethical committee.

Answer: We have agreement from the ethical committee. We will submit it in the submission system.

  1. We noted the authors need to correct the scientific names of bacteria (E coli, Psudomonas, etc) as Italic form at many palces.

Answer: We changed the names of the bacteria throughout the manuscript.

  1. This study highlighted that the number of requested blood culture tests were reduced but the positivity rates were increased. Is there any association with COVID positive patients and blood culture positive patients? Please add the possible reasons for increasing the culture tests positive at the revised manuscript.

Answer: Unfortunately, we didn't analyse SARS-CoV-2 positivity in our patients. There were only few COVID positive patients at that time. Increasing the culture tests positivity could be caused by the fact that patients were "selected"- only the most severely ill patients were admitted to the hospital due to parent's concern about COVID. 

  1. Did the authors notice co-infection cases with COVID-19 and other bacterial or viral infection cases? It will be more interesting if the authors can show co-infection cases. Please review and add at the revised one.

Answer: Unfortunately, we didn't analyse SARS-CoV-2 positivity in our patients. There were only few COVID positive children in our hospital at that time. This study focused on microbiological test changes caused by the epidemiological situation not COVID itself.

  1. The authors also described that the COVID-19 infection make great impact on adult patients than paediatrc cases. Why did the authors choose paediatric center for conducting this study?  The study from adult center will be more interesting and add the justification for choosing paediatric center at the revised manuscript.

Answer: All authors worked in a large paediatric center and experienced complications caused by SARS-CoV-2 pandemic; therefore this hospital has been chosen for the study. Moreover, we did not find any scientific reports regarding paediatric centres during the pandemic so we decided to conduct the retrospective study. 

Round 2

Reviewer 1 Report

The authors have partially improved their manuscript. Therefore, I would recommend that you focus on the following points:

- I don't see any citations for Tables 1 through 11 in the manuscript text. Are these tables part of the supplementary information? I would recommend highlighting the p-values in the figures themselves to emphasize the substantial differences, as well as noting these p-values throughout the manuscript text. Readers would be able to follow along more easily.

- I would recommend that the language be improved and that the manuscript text be free of grammatical errors once more. The authors haven't given it a serious shot yet.

Author Response

Dear Reviewer, thank you for all your suggestions. 

  1.  I don't see any citations for Tables 1 through 11 in the manuscript text. Are these tables part of the supplementary information? I would recommend highlighting the p-values in the figures themselves to emphasize the substantial differences, as well as noting these p-values throughout the manuscript text. Readers would be able to follow along more easily.

Answer: We added the citations as well as the p-values to the text. The tables are not supplementary, they should be a part of the manuscript unless the Editor decided otherwise. We cannot add the p-values to the figures because in our opinion it would not be readable to which graph they were calculated. 

2. I would recommend that the language be improved and that the manuscript text be free of grammatical errors once more. The authors haven't given it a serious shot yet.

Answer: We conducted the language editing. 

Reviewer 2 Report

I accepted this revised version.

Author Response

Thank you very much for your help!

Best regards